# Hyperoside and Quercitrin in *Houttuynia cordata* Extract Attenuate UVB-Induced Human Keratinocyte Cell Damage and Oxidative Stress via Modulation of MAPKs and Akt Signaling Pathway

**DOI:** 10.3390/antiox11020221

**Published:** 2022-01-24

**Authors:** Nattakan Charachit, Amonnat Sukhamwang, Pornngarm Dejkriengkraikul, Supachai Yodkeeree

**Affiliations:** 1Department of Biochemistry, Faculty of Medicine, Chiang Mai University, Chiang Mai 50200, Thailand; nattakan_charachit@cmu.ac.th (N.C.); amonnat_s@cmu.ac.th (A.S.); pornngarm.d@cmu.ac.th (P.D.); 2Center for Research and Development of Natural Products for Health, Chiang Mai University, Chiang Mai 50200, Thailand

**Keywords:** *Houttuynia cordata*, UVB, antioxidant, anti-apoptosis, HaCaT keratinocyte

## Abstract

Ultraviolet radiation is a major environmental harmful factor on human skin. In this paper, we investigate the potential mechanism of *Houttuynia cordata* extract on UVB-induced HaCaT keratinocyte cell death and inflammation. We found that *Houttuynia cordata* ethyl acetate extract fraction (HC-EA) protected against UVB-induced cell damage. The HPLC results indicate that quercitrin and hyperoside are the major polyphenolics in HC-EA and are responsible for providing protection against UVB-induced cell death. These responses were associated with the regulation of caspase-9 and caspase-3 activation, which rescued HaCaT cells from UVB-induced apoptosis. In addition, HC-EA, quercitrin, and hyperoside attenuated UVB-induced inflammatory mediators, including IL-6, IL-8, COX-2, and iNOS. Furthermore, the treatment of cells with HC-EA and its active compounds abolished intracellular ROS and increased levels of heme oxygenase-1 and superoxide dismutase. UVB-induced ROS production mediated Akt and mitogen activated protein kinases (MAPKs) pathways, including p38, ERK, and JNK. Our results show HC-EA, quercitrin, and hyperoside decreased UVB-induced p38 and JNK phosphorylation, while increasing ERK and Akt phosphorylation. MAPKs and Akt mediated cell survival and death were confirmed by specific inhibitors to Akt and MAPKs. Thus, HC-EA, which contains quercitrin and hyperoside, protected keratinocyte from UVB-induced oxidative damage and inflammation through the modulation of MAPKs and Akt signaling.

## 1. Introduction

Ultraviolet B (UVB) exposure is a major extrinsic cause of skin damage, aging, inflammation, tanning, wrinkling, and skin carcinogenesis. These biological effects are based on electromagnetic energy converted from UVB radiation to molecular oxygen, which in turn generates certain pivotal reactive oxygen species (ROS), such as hydrogen peroxide, superoxide anions, and hydroxyl radicals. The accumulation of ROS can also induce intracellular DNA damage by oxidation of deoxyribonucleotide bases leading to gene mutations. Moreover, UVB can also directly induce DNA damage, in particular cyclobutane pyrimidine dimers and 6-4 photo adducts, in human skin cells [1]. In order to prevent DNA mutation in skin cells, the DNA damage must be repaired, as excessive DNA damage beyond the intracellular repair capacity leads to cell death.

The direct DNA damage and ROS production caused by UVB exposure lead to the activation of cytoplasmic signal transduction pathways, which are related to cell survival, death, and inflammation [2]. However, well-functioning skin cells employ a protective antioxidant system that involve superoxide dismutase (SOD), catalase, glutathione peroxidase, and heme oxygenase-1 (HO-1) to defend against the destructive effects of ROS [3]. The loss of intracellular redox balance in the skin during UVB-mediated oxidative stress results in cellular damage and eventually leads to apoptotic cell death. A good deal of evidence indicates that the induction of keratinocyte apoptosis by UVB radiation is mediated by both the mitochondrial (intrinsic) and death receptor (extrinsic) pathways [4]. Moreover, chronic UVB irradiation increases the production of proinflammatory mediators, such as tumor necrosis factor-α, interleukin-1α (IL-1α), IL-1β, IL-6, IL-8, prostaglandin E2, and nitric oxide [5]. Therefore, it has been suggested that inflammation plays a role in the pathogenesis of skin diseases under UVB exposure.

In keratinocyte cells, UVB radiation activates PI3K/Akt and mitogen-activated protein kinases (MAPKs), including extracellular regulated kinase (ERK), c-Jun N-terminal kinase (JNK), and p38 [6,7]. These signaling pathways play roles in mediating ROS, triggering certain cellular events, including proliferation, differentiation, apoptosis, and inflammation. UVB-induced ROS has been reported to be able to activate p38 and JNK activity, which in turn induces apoptosis through the intrinsic pathway [8,9]. Moreover, the activation of p38 and JNK is known to induce proinflammatory mediators [10,11]. On the other hand, the activation of Akt and ERK can function as pro-surviving signaling against UVB-induced apoptosis [12]. Additionally, the activation of ERK also induces the expression of antioxidant enzymes [13,14]. Accordingly, identifying the efficient agents that can regulate the activation of MAPKs in conjunction with the Akt pathway is a potential first step in developing a strategy to protect against UV-induced skin damage and inflammation.

According to their safety and effectiveness, natural compounds have been of considerable interest in the field of cosmetic research and development for the prevention of and protection against UV-induced skin photodamage and inflammation. *Houttuynia cordata* Thunb (HC) is a herbal plant that is distributed throughout Japan, Korea, China, and Southeast Asian. The extracts of *H. cordata* have been shown to be therapeutic and have been used to treat various diseases, such as cancer, diabetes, obesity, skin diseases, and severe acute respiratory syndrome [15,16,17,18]. Several investigators have provided evidence of the antioxidant and anti-inflammation properties of *H. cordata* extracts and their bioactive molecules [19]. For example, ethyl acetate fractions of *H. cordata* have been shown to be able to suppress NF-κB activity and attenuate the activation of MAPKs (p38 and JNK) in LPS induced RAW 264.7 macrophage cells. Furthermore, the extracts have been found to significantly reduce NO, PGE-2, TNF-α, and IL-6 levels [20]. In addition, *H. cordata* extract inhibited the generation of ROS in TNF-α stimulated human keratinocyte cells and activated Nrf2, along with the subsequent induction of antioxidative enzymes [21,22]. *H. cordata* contains a wide range of polyphenols, such as rutin, quercetin, hyperoside, quercitrin, and chlorogenic acid, which are known to be responsible for its pharmacological activity, including its anti-inflammatory and antioxidative properties [17,19].

Even though *H. cordata* extracts have been associated with various efficacies, little is known about the protective effects of *H. cordata* extract and its bioactive compounds against UVB radiation-induced skin damage and inflammation, especially with regard to human keratinocyte. Therefore, we investigate the protective effects of *H. cordata* extract and the active compounds that can offer protection against UVB-induced damage and inflammation in keratinocyte cells.

## 2. Materials and Methods

### 2.1. Chemicals and Reagents

Dulbecco’s modified Eagle’s medium (DMEM), penicillin/streptomycin, and trypsin-EDTA were purchased from Gibco (Grand Island, NY, USA). Fetal bovine serum (FBS) was supplied by Hyclone (Logan, UT, USA). MitoView^TM^ 633 was purchased from Biotium (Fremont, CA, USA). IL-6, IL-8 ELISA kits and FITC Annexin V kit were obtained from BioLegend (San Diego, CA, USA). LY294002, PD98059, antibodies specific to Akt, β-actin, COX-2, JNK, p38, p44/42 (ERK), p-Akt, p-JNK, p-p38, and p-p44/42 (ERK) were obtained from Cell Signaling Technology (Danvers, MA, USA). Antibodies specific to Active Caspase-3, PARP1, HO-1, SOD1, and NRF2 were purchased from Abclonal (Woburn, MA, USA). The DCF-DA and antibody specific to iNOS were obtained from Sigma (St. Louis, MO, USA). Nitrocellulose membrane and ECL reagent were supplied by GE Healthcare (Little Chalfont, UK).

### 2.2. Plant Materials and Sample Preparation

*H. cordata* was provided from CMH Chiangmai Holding Co., Ltd. (Chiangmai, Thailand). A specimen herbarium voucher (No. 002360) was deposited in the Herbarium of the Pharmacy Faculty (CMU Herbarium), Chiang Mai University. The leaves were cut into pieces, air dried at 60 °C by hot air oven and powdered. A ground powder (502 g) was extracted twice with 80% *(v/v)* ethanol overnight at the room temperature. The solvent was then filtered, concentrated under vacuum evaporation, and freeze dried to obtain the ethanolic fraction. The ethanolic fraction (HC-ET) was partitioned with hexane to remove pesticides from the extract, partitioned with dichloromethane and ethyl acetate, then concentrated under vacuum evaporation, and air dried to obtain the dichloromethane fraction (HC-DC) and the ethyl acetate fraction (HC-EA), respectively.

### 2.3. Total Phenolic Contents

Total phenolic contents were measured by the modified Folin–Ciocalteu method [23]. The DMSO-dissolved extract fractions were mixed with Folin–Ciocalteu reagent and incubated at room temperature for 3 min. Then, sodium carbonate (Na_2_CO_3_) was added to the samples and incubated for 30 min. The absorbance was measured after incubation at 765 nm by spectrophotometry and compared with standard gallic acid (GA).

### 2.4. Total Flavonoid Contents

Total flavonoid contents were measured by the aluminum chloride spectrophotometric assay as previously described [24]. The extract fractions were dissolved in DMSO and mixed with sodium nitrite (NaNO_2_). After incubation in room temperature for 5 min, the samples were added with aluminum chloride (AlCl_3_) and incubated for 5 min. Then, the samples were incubated with sodium hydroxide (NaOH) for 15 min. After incubation, the absorbance was measured at 510 nm by spectrophotometry and compared with standard catechin.

### 2.5. High-Performance Liquid Chromatography (HPLC) Analysis

*H. cordata* extract fractions were determined the components by HPLC using an Eclipse Plus C18 (5 μm, 4.6 × 250 mm, Agilent, Santa Clara, CA, USA). Samples were dissolved in ethanol and injected 10 μL to HPLC with UV detection (HPLC-UV). The mobile phases were 0.1% *(v/v)* trifluoroacetic acid (TFA) in water as solvent A and methanol as solvent B in the gradient elution program with flow rate of 1 mL/min. The samples were detected for fingerprint analysis and compared with standard of compounds found in *H. cordata*, such as chlorogenic acid, hyperoside, quercitrin, and quercetin. The contents of each phenolic compound were calculated by the HPLC peak area under the curve, compared with the standard calibration curve.

### 2.6. DPPH (2,2-Diphenyl-1-picrylhydrazyl) Assay

The free radical scavenging ability of the extracts was tested by DPPH radical scavenging assay with slight modifications, which determined the hydrogen atom donating ability of the extracts by the decolorization of methanol solution of DPPH [25]. In a methanol solution, DPPH generated a violet/purple color and faded to shades of the yellow color in the presence of antioxidants. A solution of 0.1 mM DPPH in methanol was prepared, and 20 μL of several concentrations of the extracts or vitamin E was mixed with 180 μL of DPPH solution in a 96-well plate. After 20 min of incubation in the dark, the absorbance was measured at 520 nm using a microplate reader.

### 2.7. ABTS (2,2′-Azino-bis(Ethylbenzthiazoline-6-sulfonic acid)) Assay

The ABTS radical scavenging ability was performed as in the previous study with slight modifications [26]. ABTS^•+^ cation radical was generated by a reaction between 7 mM ABTS and 4.9 mM potassium persulfate (K_2_S_2_O_8_) (1:1, *v/v*), incubated at room temperature for 12–16 h in the dark before use. To achieve an absorbance of 0.700–0.720 at 734 nm, ABTS^•+^ solution was diluted with DI water. A total of 10 μL of various concentrations of HC-EA, standard compounds or trolox was mixed with 990 μL of diluted ABTS^•+^ solution, then incubates in the dark for 6 min. After incubation, the absorbance was measured at 734 nm by spectrophotometry. All the measurements were performed in duplicate.

### 2.8. Cell and Cell Culture

Human keratinocyte cells (HaCaT cells), provided by the American Type Culture Collection (ATCC, Manassas, VA, USA), were cultured in DMEM supplemented with 10% FBS and 1% penicillin/streptomycin at 37 °C in a humidified 5% CO_2_ incubator.

### 2.9. Sample Treatment and UVB Irradiation

HaCaT cells were seeded at a density of 7.5 × 10^4^ cells/well in 24-well plates overnight, then pretreated with different concentrations of HC-EA or active compounds in 0.5% FBS DMEM for 6 h prior to UVB irradiation. The cells were washed once with 250 μL of PBS, added 250 μL of PBS supplemented with the HC-EA or standard compounds then irradiated with UVB 15 mJ/cm^2^ using UVB generator (UVP CL-1000 Ultraviolet Crosslinker, Upland, CA, USA) and further incubated with 0.5% FBS DMEM supplemented with HC-EA or standard compounds for 1–24 h.

### 2.10. Cell Viability

Cell viability was measured by a sulforhodamine B (SRB) assay [27]. After an incubation period, the cells were washed once with sterilized PBS then fixed with 250 μL of 10% *(w/v)* trichloroacetic acid (TCA) to each well and incubated at 4 °C for 1 h. Then, the plates were washed with slow-running tap water, the excess water was removed using paper towels and left to air dry at room temperature (20–25 °C). The cells were stained for 30 min by adding 250 μL of 0.057% *(w/v)* SRB solution to each well. The unbound dye was removed by washing with 1% *(v/v)* acetic acid. The protein-bound dye was dissolved in 250 μL of 10 mM Tris base solution and transferred 100 μL to 96-well plate to measure the OD at 510 nm in a microplate reader.

### 2.11. ROS Generation

Intracellular ROS was analyzed by 2′,7′-dichlorodihydrofluorescein-diacetate (DCF-DA assay) [28]. DCF-DA enters the cell by passive diffusion to the cytosol and is deacetylated by cellular esterases to from non-fluorescent derivative 2′,7′-dichlorofluorescein (DCF). Upon exposure to ROS as well as other oxidants, such as reactive nitrogen species, DCF undergoes two-electron oxidation, resulting in the generation of the highly fluorescent DCF. Since then, there have been numerous reports using DCF-DA to measure intracellular ROS in UVB-induced HaCaT cells [29,30]. Briefly, HaCaT cells were pretreated for 6 h prior UVB irradiation then incubated for 2 h. After incubation, HaCaT cells were stained with 5 μM of DCF-DA for 30 min at 37 °C, and then the cells were washed twice with PBS and lysed by 90% of DMSO in PBS. The fluorescence was measured at excitation/emission of 485/525 nm by fluorescence microplate reader.

### 2.12. Enzyme-Linked Immunosorbent Assay (ELISA)

The culture supernatants from HaCaT cells were collected at 24 h after UVB irradiation. The production of IL-6 and IL-8 in the culture medium were quantified by ELISA kits (Biolegend, San Diego, CA, USA), according to the manufacturer’s instructions. Absorbance was read at 450 nm by microplate reader.

### 2.13. Apoptosis Evaluation 

The analysis was carried out using a commercial FITC Annexin V kit (BioLegend, San Diego, CA, USA. Cat. No. 420201). The cells were collected using mild trypsinization, including the dead floating cells. After washing the collected cells with PBS, the cells were stained with FITC Annexin V and Propidium Iodide for 20 min according to the manufacturer’s protocol. The stained cells were analyzed using the flow cytometer (Backman Coulter DxFLEX) and data analysis was performed using the CytExpert for DxFLEX software.

### 2.14. Mitochondrial Membrane Potential

Mitochondrial membrane potential was assessed using MitoView^TM^ 633 (Biotium, Fremont, CA, USA), according to the manufacturer’s protocol. Briefly, the cells were harvested by trypsinization at 6 h after UVB irradiation, then incubated with 100 nM of MitoView^TM^ 633 in incomplete DMEM for 20 min in the dark at 37 °C in a humidified atmosphere containing 5% CO_2_. Then, the cells were washed with PBS and fluorescence was recorded at excitation/emission of 638/660 nm by flow cytometry. Data were analyzed using CytExpert for DxFLEX software.

### 2.15. Western Blot Analysis

After incubation, the cells were collected using trypsinization and washed with PBS. The cell pellet was lysed using RIPA buffer containing protease inhibitors for 20 min on ice. The protein concentration was quantified using the Bradford method (Bio-Rad Laboratories, Des Plaines, IL, USA). In order to determine the expression level of protein in the whole cell lysate, cytoplasm and nuclear faction were separated by the SDS-PAGE electrophoresis and transferred to the nitrocellulose membrane by electroblotting. The membrane was blocked with 5% skim milk in TBS containing 0.3% *(v/v)* Tween-20 for 1 h and then incubated with primary antibodies at 4 °C overnight. The membrane was incubated with a secondary antibody for 2 h and detected by chemiluminescence.

### 2.16. Statistical Analysis

All data from three independent experiments were presented as mean ± SD. Statistical analysis was performed by one-way ANOVA with Dunnett’s test using IBM^®^ SPSS^®^ Statistics V.28.0.1.0 (142). Statistical significance was considered at *p*-value less than 0.05.

## 3. Results

### 3.1. Ethyl Acetate Extract Fractions (HC-EA) Attenuate UVB-Induced HaCaT Cells Death

The effect of *H. cordata* extract fractions on cell viability of human keratinocyte cells were determined by SRB assay. As is shown in Figure 1A, treatment of the cells with HC-ET and HC-EA at a high concentration (200 μg/mL) for 24 h had no effect on cell viability. On the other hand, treatment of the cells with HC-DM below 50 μg/mL revealed no cytotoxicity in HaCaT cells (>80% cell survival). Therefore, *H. cordata* extract fractions at 50 µM were used in further experiments. To examine the effects of *H. cordata* extract fractions on cell viability following UVB irradiation, HaCaT cells were pre-treated with 50 μg/mL of *H. cordata* extract fractions, followed by exposure to UVB irradiation (15 mJ/cm^2^). As is shown in Figure 1B, UVB exposure markedly reduced the viability of HaCaT cells to 57.1% when compared to non-exposed cells. The pretreatment of cells with 50 μg/mL of HC-EA significantly increased cell viability to 82.0% in HaCaT cells exposed to UVB. However, HC-ET and HC-DM at 50 μg/mL did not protect cells from UVB-induced cell death. Moreover, pretreatment of the cells with different concentrations of HC-EA (0–100 μg/mL) induced cell viability in a concentration-dependent manner (Figure 1C). Consequently, the resulting data indicate that HC-EA can effectively enhance cell viability in the presence of UVB.

### 3.2. Quercitrin and Hyperoside in HC-EA Protect HaCaT Cells from UVB-Induced Cell Death

The phytochemical contents in HC-EA fraction were investigated. As is shown in Table 1, HC-EA contained high levels of total phenolics and total flavonoids with values of 781.7 and 437.1 mg/g extract, respectively. Polyphenols, such as quercetin, quercitrin, hyperoside, and chlorogenic acid, are the major flavonoids and phenolics in *H. cordata*. Therefore, the major phenolic compounds in the HC-EA fraction were investigated by HPLC. The chromatogram of the HC-EA fraction exhibited peaks with the same retention times as the following standard phenolics and flavonoids: chlorogenic acid (19.6 min), hyperoside (29.1 min), quercitrin (31.0 min), and quercetin (34.0), as is shown in Appendix A. The quantities of hyperoside, quercitrin, quercetin, and chlorogenic acid in HC-EA were 230.8, 286.9, 7.8, and 63.4 mg/g extract, respectively (Table 1). Subsequently, we investigated the effects of the main phytochemicals in the HC-EA fraction, hyperoside, and quercitrin, on UVB-induced HaCaT cell death. As is shown in Appendix A, the treatment of cells with quercitrin, hyperoside, and chlorogenic acid at 100 μM had no effect on cell cytotoxicity. However, quercitrin and hyperoside treatment protected the cells from the toxic effects of UVB irradiation in a dose-dependent manner, whereas chlorogenic acid had no effect (Figure 2B–D). The resulting data indicate that hyperoside and quercitrin are the major active compounds that serve as potential barriers to UVB-induced HaCaT cell death in HC-EA fractions.

### 3.3. Effect of HC-EA and Its Active Compounds on UVB-Induced HaCaT Cell Apoptosis

We investigated whether HC-EA, quercitrin, and hyperoside protected HaCaT keratinocytes from UVB-induced apoptosis. HaCaT cells were pre-treated with HC-EA (0–100 μg/mL), quercitrin (100 μM), or hyperoside (100 μM) before being irradiated with UVB, while apoptosis was assayed by annexin V and PI staining. As expected, UVB irradiation induced HaCaT cell apoptosis by 30.1%, whereas pretreatment with HC-EA significantly decreased UVB-induced HaCaT cell apoptosis in a dose-dependent manner. In addition, quercitrin and hyperoside at 100 μM also decreased the cell apoptosis to 18.2 and 18.3%, respectively (Figure 3A,B). UVB irradiation activates the intrinsic death effector pathway that perturbs mitochondrial structure and function leading to cell apoptosis. Mitochondrial apoptosis is initiated by an alteration of mitochondrial membrane potential (MMP). This causes permeability transition pores to be opened causing the discharge of caspase activators, which in turn induces apoptosis. To investigate MMP alteration in incidences of UVB irradiation, MitoView^TM^ 633 stain was used. The dye is membrane permeant and becomes brightly fluorescent upon accumulation in the mitochondrial membrane. The changes of the MMP were analyzed by flow cytometry. As is shown in Figure 3C,D, HaCaT cells were radiated by UVB treatment to initiate disruption of MMP by around 28.0%. However, pretreatment with HC-EA with 50 μg/mL and 100 μg/mL before UVB radiation significantly reduced the loss of MMP to 21.8 and 16.3%, respectively. In addition, pretreatment with quercitrin and hyperoside at 100 μM also protected cells from UVB-induced cell death against a loss of MMP to 20.6 and 20.2%, respectively.

The involvement of the caspase pathway in UVB-induced apoptosis has previously been documented. The effect of UVB radiation on the activation of caspases in HaCaT cells was determined using Western blot analysis. As is shown in Figure 3E,F, a significant upregulation of cleaved caspase-3 and caspase-9 was observed in the UVB irradiated group when compared with the control. However, HC-EA at 50 and 100 μg/mL, or quercitrin and hyperoside at 100 μM, inhibited the UVB-induced cleavage of capase-3 and caspase-9. PARP-1 is known to be a downstream substrate of caspase-9 and -3. UVB radiation occurred in the cellular levels of cleaved PARP-1 in HaCaT cells, while the pretreatment of the cells with HC-EA, quercitrin, or hyperoside decreased the UVB-induced cleavage of PARP-1. Collectively, the resulting data would indicate that HC-EA, quercitrin, and hyperoside exhibited a protective effect on UVB-induced apoptosis via the suppression of caspase-3 and-9 activation.

### 3.4. Anti-Inflammatory Effect of HC-EA and Its Active Compounds on UVB-Irradiated HaCaT Cells

To analyze whether HC-EA, quercitrin, and hyperoside could inhibit UVB-induced proinflammatory mediators in HaCaT cells, the expression level of proinflammatory cytokines, including IL-6 and IL-8 along with certain proinflammatory enzymes, such as iNOS and COX-2, were investigated. As is shown in Figure 4A–C, pretreatment of the cells with HC-EA, quercitrin, and hyperoside decreased UVB-induced IL-6 in a concentration-dependent manner. In addition, the level of IL-8 increased after UVB-irradiation and pretreatment with HC-EA and quercitrin, and hyperoside, which then reduced IL-8 levels in a dose-dependent manner (Figure 4D–F). The expression levels of iNOS and COX-2 were investigated using Western blot analysis. As is shown in Figure 4G,H, pretreatment of HC-EA, quercitrin, and hyperoside declined a UVB-induced effect through the expression levels of iNOS and COX-2. These results demonstrated that HC-EA, quercitrin, and hyperoside effectively suppressed UVB-induced inflammation via reduced proinflammatory enzyme and cytokine contents.

### 3.5. Scavenging Activity of HC-EA and Its Active Compounds against UVB-Generated ROS in HaCaT Cells

UVB induced the overproduction of ROS, which then induced the production of the proinflammatory mediator and triggered keratinocyte apoptosis. Therefore, intracellular ROS generation was monitored to investigate the effects of HC-EA, quercitrin, and hyperoside on UVB-induced oxidative stress in HaCaT cells. The generation of intracellular ROS was detected using a DCF-DA fluorescent probe. As is shown in Figure 5A, the fluorescent intensity in UVB-irradiated cells produced a significantly greater increase (161.2%) relative to non-UVB-irradiated cells (100%). However, the treatment of the cells with HC-EA at 50 and 100 μg/mL potently reduced UVB-induced intracellular ROS to 115.1 and 111.0%, respectively, whereas the pretreatment of the cells with NAC at 1 and 2 mM also significantly reduced UVB-induced intracellular ROS production. Furthermore, quercitrin and hyperoside at 50 and 100 μM significantly declined UVB-induced ROS production in HaCaT cells. We then investigated the effect of HC-EA and its active compounds on the expression level of antioxidant enzymes, including SOD1 and HO-1. HC-EA significantly increased the expression levels of SOD1 and HO-1. The treatment of the cells with quercitrin at 100 μM induced the protein expression capabilities of SOD1 and HO-1 (Figure 5B,C). Moreover, the effect of HC-EA and its active compounds on the expression level of Nrf2, the main transcription factor responsible for oxidative stress and cytoprotecting, was investigated. As is shown in Figure 5C,D, HC-EA, quercitrin and hyperoside treatment consistently enhanced the expression levels of Nrf2 in UVB-induced HaCaT cells. The radical scavenging activities of HC-EA and its active compounds were further evaluated in terms of their free radical scavenging properties by DPPH and ABTH assays. As is shown in Table 2, the inhibition concentrations of 50% (IC_50_) of HC-EA, quercitrin, and hyperoside on DPPH were 21.7 μg/mL, 70.7 μM, and 55.3 μM, respectively, whereas vitamin E was 36.1 μg/mL. The antioxidant effects of HC-EA, quercitrin, and hyperoside on the de-coloration of ABTS were exhibited as IC_50_ at 4.5 μg/mL, 13.3 μM, and 12.2 μM, respectively. Alternatively, the positive control, Trolox, exhibited IC_50_ at 2.4 μg/mL. These findings demonstrate that HC-EA and its active compounds have the capacity to reduce the generation of ROS, either through its own ROS scavenging ability or by enhancing intracellular antioxidant enzymes.

### 3.6. Modulation of MAPKs and Akt Signaling Pathway by HC-EA and Its Active Compounds in UVB-Irradiated HaCaT Cells

UVB-activated MAPKs and Akt signaling pathways have been determined to be involved in skin aging. Therefore, the effects of HC-EA and its active compounds on UVB-induced activation of Akt and MAPKs, including ERK, JNK, and p38, were investigated. As is shown in Figure 6A–D, UVB stimulated the phosphorylation of ERK1/2, JNK, and p38. For the HC-EA, quercitrin and hyperoside treatment of UVB-exposed cells, the levels of phosphorylated p-38 and JNK were reduced when compared to UVB-exposed cells. In contrast, the phosphorylation level of ERK was increased when the cells were pretreated with HC-EA, quercitrin, and hyperoside. In order to verify that the activation of ERK by HC-EA and its active compounds indeed had a promising effect on photoprotective in UVB-induced cell death, HaCaT cells were cotreated with PD98056 (ERK inhibitor) and HC-EA, or its active compounds, before UVB irradiation. Subsequently, cell survival was then determined. A combination treatment with PD98056 and HC-EA, or its active compounds, significantly reduced cell viability when compared with HC-EA or its active compounds alone (Figure 6E). This result indicated that the ERK inhibitor attenuated the protective effect of HC-EA and its active compounds on UVB-induced cell death. Furthermore, HC-EA, quercitrin, and hyperoside significantly increased the phosphorylation level of Akt when compared with UVB-irradiated cells (Figure 6F,G). By combining the treatment of LY294002 (Akt inhibitor) with HC-EA or its active compounds, cell survival was found to be significantly decreased when compared with HC-EA and its active compounds alone, according to the pattern observed in the UVB-irradiation group (Figure 6H). Taken together, these results indicate that HC-EA, quercitrin, and hyperoside exert photoprotective effects by modulating the MAPKs and Akt signaling pathways.

## 4. Discussion

The excessive exposure to UVB can lead to a range of skin disorders, including skin aging, inflammation, hyperplasia, and skin cancer. UVB radiation generates ROS in a variety of skin cells, including keratinocyte. The oxidative stress generated by ROS can promote cell apoptosis, as well as change certain biological macromolecules, including DNA, proteins, and the lipid membrane, while also activating the inflammatory signaling pathway [31]. Therefore, diminishing the oxidative stress of skin cells is one of the main strategies for the prevention of UVB-induced skin damage. The use of phytochemical antioxidants, in accordance with their biocompatibility and additional health benefits, has been determined to be a favorable approach in recent studies [32,33]. Our previous study reported that *H. cordata* extract, and its active flavonoids and hyperosides, markedly decreased UVB-induced inflammation and oxidative stress in human skin fibroblast cells [34]. However, the potential benefits of *H. cordata* extract and its active compounds on UVB-induced keratinocyte injuries, including ROS production, apoptosis, and inflammation, have not yet been fully investigated. In this study, we found that HC-EA fractions exhibited protection against UVB-induced cell death.

The antioxidant effects of *H. Cordata* extract may markedly contribute to the beneficial properties of phenolics and flavonoids. The phytochemicals in *H. cordata*, such as chlorogenic acid, quercetin, hyperoside, quercitrin, kaempferol, and rutin, have been found to exhibit antioxidant capacities [19]. However, the main active polyphenolic compounds responsible for this photoprotective effect in *H. cordata* have not yet been investigated. Our results demonstrated that HC-EA was rich in phenolics and flavonoids, while the main polyphenolic components in HC-EA were quercitrin and hyperoside. This determination is supported by the findings of the study conducted by Tian et al., who demonstrated that the ethyl acetate fraction derived from *H. cordata* was rich in quercitrin and hyperoside, followed by chlorogenic acid, all of which were responsible for the existing antioxidant properties [35]. It is rational to speculate that the main polyphenolic compounds in HC-EA might contribute to offer protection against UVB-induced keratinocyte cell death. In this study, we found that quercitrin and hyperoside treatment protected the cells from the toxic effects of UVB irradiation, while chlorogenic acid had no effect. This result was similar with that of previous studies, which reported that quercetin, the non-glycoside form of quercitrin and hyperoside, is a potential agent against UVB irradiation-induced skin damage [36].

UVB radiation triggers skin cell damage, which can then lead to a loss of keratinocytes or the induction of keratinocyte apoptosis. Remarkably, the agents that reduce UVB-induced apoptosis could also have a beneficial effect on photoprotection. Our study indicated that HC-EA, quercitrin, and hyperoside could prevent UVB-induced HaCaT cell apoptosis. UVB radiation is known to trigger the activation of both the intrinsic and extrinsic apoptotic signaling pathways [37]. However, evidence suggests that the UVB-induced apoptosis of keratinocyte is primarily mediated via the intrinsic pathway or the mitochondrial death signaling pathway [4]. Mitochondria are both the major producers and the target of ROS in human cells. UVB-induced ROS can damage mitochondria and result in mitochondrial structure change and disruption of MMP, which then leads to the insertion of proapoptotic proteins into the membrane. This could then create the mitochondrial permeability transition pore, thereby releasing the cytochrome c into the cytoplasm, which in turn triggers other downstream events in the apoptotic cascade [38]. In line with the outcomes of a previous study, we also observed that UVB-irradiation induced the disruption of MMP in HaCaT cells. However, pretreatment of the cells with HC-EA, quercitrin, and hyperoside rescued UVB-induced MMP loss. PARP-1 can act as highly sensitive sensor for DNA damage and plays a key role in DNA repair resulting from oxidative stress. The activation of PARP-1 facilitates cellular disassembly and serves as a marker of the cells undergoing apoptosis. Caspase-3 plays a central role in the execution of apoptosis and is primarily responsible for the activation of PARP during cell death [39]. Consistent with the apoptosis assay, our results indicate that pretreatment of the cells with HC-EA, quercitrin, and hyperoside could effectively reduce the UVB-induced expression levels of cleavage-PARP and cleavage-caspase-3. UV radiation predominantly activates the intrinsic apoptotic pathway, which is induced by a collapse of the mitochondrial membrane potential and results in activation of procaspase 9. Active caspase-9 initiates a caspase activation cascade, leading to proteolytic procaspase-3 and-7 activation. The latter caspases then activate other caspases as well as cleaving structural and regulatory proteins, which then result in the full-blown apoptotic phenotype [40]. IN this study, we found that HC-EA, quercitrin, and hyperoside decreased UVB-induced caspase-9 activation, which in part accounts for its protective effect of the UVB-mediated disruption of its mitochondrial membrane potential. Taken together, these results suggest that HC-EA, quercitrin, and hyperoside could prevent UVB-induced keratinocyte cell apoptosis via the regulation of caspase-3 and caspase-9 activation and regulating the mitochondrial apoptotic pathway.

Keratinocyte cells are a source of certain UVB-induced inflammatory mediators, such as cytokines and chemokines [41]. The production of ROS in significant amounts arises from UVB-induced oxidative stress-initiated inflammation-mediated skin aging. UVB-mediated skin damage is correlated with the production of proinflammatory cytokines and certain inflammatory enzymes, such as TNF-α, IL-6, IL-8, COX-2, and iNOS [42]. In accordance with our experimental data, we have confirmed that the expression levels of IL-6, IL-8, COX-2, and iNOS were markedly increased after UVB irradiation in HaCaT cells and were successfully inhibited by treatment with HC-EA. Moreover, the main active compounds in HC-EA, quercitrin and hyperoside, also downregulated proinflammatory cytokines and inflammatory enzymes in UVB-irradiated cells. In accordance with the findings of a study conducted by Yin et al., it has been demonstrated that quercitrin reduced the inflammatory response in the superficial and deep dermis of mice skin that had been induced by UVB irradiation [36]. Altogether, HC-EA inhibited the key components of an inflammatory response in HaCaT cells. It can therefore be postulated that the main active components of HC-EA, including hyperoside and quercitrin, are accountable for reducing the proinflammatory mediators.

UVB promotes oxidative stress by worsening the production of certain ROS, such as superoxide radicals, hydroxyl radicals, and hydrogen peroxide, while decreasing the components of the endogenous antioxidant system including catalase, SOD-1 and HO-1, and by reducing glutathione [43]. These are very lethal and can cause extensive damage to biomolecules in skin cells leading to skin damage and inflammation. Therefore, the use of an antioxidant agent can prevent the skin damage caused by UVB irradiation. The DCF-DA assay revealed that HC-EA, quercitrin, and hyperoside effectively reduced the intracellular ROS generated by UVB irradiation. Keratinocytes are more sensitive to oxidative stress than skin fibroblasts because they are lower in terms of first-line antioxidant enzymes activity [44]. The induction of enzymatic antioxidants could be a defense mechanism used to reduced ROS against oxidative stress. In our study, UVB slightly induced HO-1 and SOD-1 expression in HaCaT cells, which is consistent with several previous reports that found that the expression level of antioxidant enzymes is induced by exposure to moderate oxidative stress [45,46]. Interestingly, HC-EA, quercitrin, and hyperoside treatment could increase the expression levels of HO-1 and SOD-1, thereby contributing to an improvement in the redox balance in UVB-induced cells. Nrf2 has been suggested to play a protective role in UVB-mediated oxidative stress. Nrf2 regulates the expression of antioxidant enzymes including SOD, HO-1, catalase, and glutathione peroxidase [45,46]. Therefore, we determined the activity of HC-EA and its active compounds on the expression level of Nrf2 in UVB-irradiated cells. Our result shown that HC-EA, quercitrin, and hyperoside induced the expression level of Nrf2. The best characterized mechanism of antioxidant action of flavonoids is due to their ability to interact with ROS by scavenging or reducing them. From the previous studies, the ethyl acetate fraction of *H. cordata*, hyperoside, and quercitrin exhibited high reducing capacities [35,47,48]. The reduction property of the substance has a close correlation with its antioxidant capacity by donating electron or a hydrogen atom. In this study, the free radical scavenging properties of HC-EA and its active compounds were investigated using the DPPH and ABTS assays. These assays evaluated the ability of the compound to donate a hydrogen to DPPH and to transfer an electron to ABTS^+^. Our results indicated that HC-EA, quercitrin, and hyperoside presented free radical scavenging activity in both methods, confirming the previous reports on their antioxidant potential.

UVB exposure induces ROS, which in turn activates several cellular signaling events. The Akt and MAPK signaling pathways, including JNK, ERK1/2, and p38, play a pivotal role in the response to UVB-induced ROS generation that mediates the actions associated with skin inflammatory responses and apoptosis. The activation of the p38 and JNK signaling pathways are associated with the initiation of UVB-induced apoptosis [49,50]. The phosphorylation of p38 and JNK induced the cytochrome c release from the mitochondria and an induction of caspase-9 and capase-3 activation, resulting in apoptosis through the intrinsic pathway. In addition, several studies have reported that the activation of p38 and JNK can trigger inflammatory mediators in keratinocyte after UVB exposure [12]. Our results demonstrated that UVB-induced p38 and JNK activation in HaCaT cells, and the treatment of the cells with HC-EA, quercitrin, and hyperoside, can attenuate the phosphorylation of p38 and JNK with a marked decrease mainly in p38. This suggests that HC-EA and its active compounds reduced UVB-induced cells apoptosis and inflammation via the inhibited p38 and JNK pathway. On the other hand, the stimulation of the ERK and PI3K/Akt signaling pathways plays an important role in regulating cell survival and proliferation [51]. Some plant flavonoids exhibited anti-apoptotic properties by activating the ERK and Akt signaling pathways [52]. In this study, we found that the levels of phosphorylated ERK and Akt were increased after the treatment with HC-EA, quercitrin, and hyperoside in UVB-irradiated cells. To confirm the role of ERK and Akt in HC-EA, and that its active compounds prevented UVB-induced cell death, the inhibitors of ERK and Akt were used. The obtained results indicated that, when the HaCaT cells were pretreated with ERK or Akt inhibitors together with HC-EA or its active compounds, a reversal was observed in the protective effect of HC-EA, quercitrin, and hyperoside in UVB-induced cell death. Moreover, several studies have reported that Nrf2 activation is facilitated by ERK and Akt pathway [53]. The activation of Nrf2 leads to the induction of the expression of antioxidant enzymes, including SOD and HO-1. Based on what was mentioned above, we suggest that the Nrf2 activation induced by HC-EA and its active compounds may be dependent on the activation of the ERK and Akt signaling pathway. This would suggest that the activation of ERK and Akt signaling by HC-EA and its active compounds is crucial for cellular homeostasis and cellular adaptation, promoting HaCaT cell survival and inhibiting UVB-induced cell death. 

Interestingly, the above results indicate that quercitrin and hyperoside at 100 μM reduced UVB-induced HaCaT cell apoptosis from 30.1 to 18.2 and 18.3%, respectively, whereas HC-EA at 50 μg/mL reduced the cells apoptosis to 17.4 %. In addition, antioxidant effect of quercitrin and hyperoside at 100 μM demonstrated equivalent efficacy to HC-EA at 50 μg/mL. According to the amount of quercitrin and hyperoside in HC-EA were 286.9 and 230.8 mg/g extract, respectively. Therefore, the concentration of quercitrin and hyperoside in 100 μg/mL HC-EA is equal to 63.7 and 49.5 μM. This could imply that the cytoprotective effect of HC-EA in UVB-induced HaCaT cells is the additive effect between quercitrin and hyperoside.

Topical application is the classical route of antioxidant compound administration to the skin. However, administration efficiency can be achieved depending on the stability and ability of compounds to penetrate to the skin and be present in their active form. Polyphenolic biotransformation is much less studied in the skin than in other organs, such as liver, kidney, and intestine. Keratinocytes contain a vast majority of xenobiotic metabolizing enzymes compared to other skin cells [54]. Topically applied dermatological skin care products containing polyphenolics may be activated by phase I and phase II enzymes in the skin. The expression of phase II enzymes, such as glutathione S-transferases, sulfotransferase, and UDP-glucuronosyltransferase, was induced by polyphenolic compounds through the Nrf2/Keap1 signaling pathway [55], as in the study of Murakami et.al, which indicated that HaCaT cells metabolized resveratrol into resveratrol-3-O-glucuronide [56]. Recently, the systemic protection of the skin by polyphenolic compounds has been reported [57]. While polyphenols are associated with a beneficial effect on skin, this action in skin cells might take place directly via parent compounds or its metabolites. Indeed, in an in vitro study, the aglycone form of hesperidin is efficiently taken up by skin fibroblasts, but does not protect them against UVA-induced damage, while hesperetin-7-glucuronide could not be detected in skin fibroblasts as an aglycone, but it is protective against UVA radiation [58]. In contrast, quercetin and its major metabolites, quercetin-3-glucuronide, exhibited a cytoprotective activity on UVB-induced keratinocyte cell damage [59]. Therefore, the biotransformation and active metabolites of quercitrin and hyperoside in keratinocytes need to be further investigated.

## 5. Conclusions

We demonstrated for the first time that HC-EA, quercitrin, and hyperoside have the capacity to protect human keratinocytes from UVB-induced apoptosis and inflammation. These effects have been associated with oxidative stress inhibition by decreasing ROS production and the regeneration of endogenous antioxidant defenses. The anti-skin photoaging effect of HC-EA and its active compounds were modulated via the MAPKs and Akt signaling pathways. Specifically, HC-EA and its active compounds reduced the UVB-regulated phosphorylation of p38 and JNK signaling and stimulated UVB-mediated ERK1/2 and Akt activation. Therefore, the incorporation of HC-EA, which contains significant amounts of quercitrin and hyperoside, into topical delivery systems could modulate UVB-induced oxidative damage and inflammation. This would be indicative of a strong potential to establish a promising strategy to prevent skin photoaging.

## Figures and Tables

**Figure 1 antioxidants-11-00221-f001:**
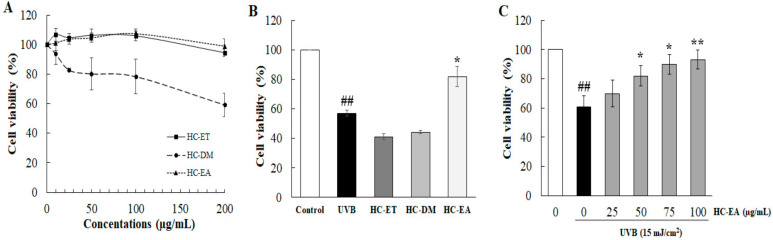
Effects of *H. cordata* extract fractions on cell viability in UVB-irradiated HaCaT cells. (**A**) HaCaT cells were treated with various concentrations of *H. cordata* extract fractions for 48 h and the cell viability was measured by SRB assay. (**B**) The protective effects of *H. cordata* extract fractions on UVB-induced HaCaT cell death. The cells were pretreated with the fractions for 6 h prior UVB irradiation (15 mJ/cm^2^), then cell viability was measured at 24 h after UVB irradiation. (**C**) Effects of HC-EA fraction on UVB-irradiated HaCaT cells. The cells were pretreated with various concentrations of HC-EA for 6 h prior UVB irradiation (15 mJ/cm^2^), then the cell viability was measured at 24 h after UVB irradiation. * *p* < 0.05, ** *p* < 0.01, as compared to the UVB-irradiated alone group. ## *p* < 0.01 as compared to the non-UV group. Data are representative of three independent experiments as mean ± SD.

**Figure 2 antioxidants-11-00221-f002:**
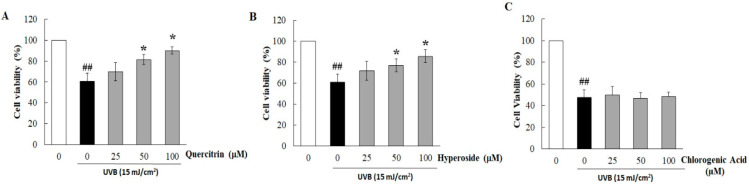
Cytoprotective effects of major compounds in HC-EA on UVB-induced cytotoxicity in HaCaT cells. The cells were pretreated with various concentrations of quercitrin (**A**), hyperoside (**B**), and chlorogenic acid (**C**) for 6 h before UVB-irradiation after 24 h the cells viability was measured by SRB assay. * *p* < 0.05 as compared to the UVB-irradiated alone group. ## *p* < 0.01 as compared to the non-UV group. Data are representative of three independent experiments as mean ± SD.

**Figure 3 antioxidants-11-00221-f003:**
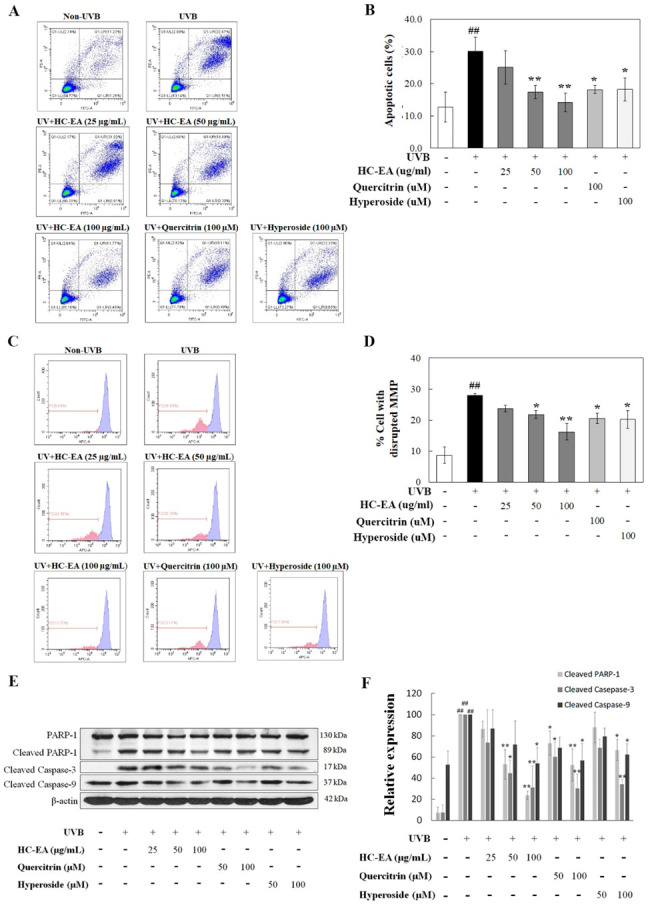
Anti-apoptotic effects of HC-EA and its active compounds on UVB-irradiated HaCaT cells. Flow cytometric analysis of apoptosis with representative histograms (**A**) and quantification (**B**) of apoptotic cells were performed to study the effects of HC-EA and its active compounds on UVB-irradiated HaCaT cells. HaCaT cells were pretreated with the indicated concentration of HC-EA, quercitrin, or hyperoside prior UVB irradiation (15 mJ/cm^2^), then the apoptotic populations were detected by FITC Annexin V/PI flow cytometry after 8 h of exposure. The effect of HC-EA and its active compounds on mitochondria membrane potential were analyzed by flow cytometer. The mitochondrial membrane potential was stained with the fluorescent dye MitoView^TM^ 633 after post-irradiation at 6 h and the result was represented in histogram (**C**) and the bar graph (**D**). The effects of HC-EA and its bioactive compounds on the expression of proteins involved in apoptosis on UVB-irradiated HaCaT cells were detected by Western blot analysis (**E**). Densitometric and statistical analysis of the apoptotic protein expression levels of cleaved PARP-1, cleaved caspase-3, and cleaved caspase-9 normalized to β-actin were performed (**F**). The experiments were repeated at least three times. * *p* < 0.05, ** *p* < 0.01, as compared to the UVB-irradiated group. ## *p* < 0.01 as compared to the control group. Data are representative of three independent experiments as mean ± SD.

**Figure 4 antioxidants-11-00221-f004:**
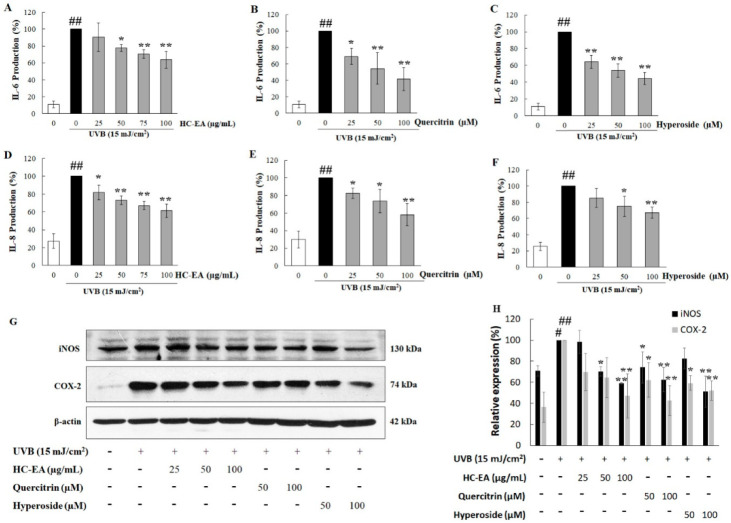
Anti-inflammatory effects of HC-EA and its active compounds on UVB-irradiated HaCaT cells. The cells were treated with various concentrations of HC-EA, quercitrin, and hyperoside or 4 h before being UVB irradiated; the culture supernatants were collected at 24 h after UVB exposure and the effects of HC-EA (**A**), quercitrin (**B**), and hyperoside (**C**) were measured on production of IL-6 (**A**–**C**) and IL-8 (**D**–**F**) using ELISA. Western blotting was performed to evaluate the effects of HC-EA and its bioactive compounds on the UVB-induced expression of proteins involved in inflammation (iNOS and COX-2) at 6 h after UVB irradiation in HaCaT cells (**G**). Densitometric and statistical analysis were performed for relative expression levels of proteins normalized to β-actin (**H**). The experiments were repeated at least three times. * *p* < 0.05, ** *p* < 0.01, as compared to the UVB-irradiated alone group. # *p* < 0.05, ## *p* < 0.01, as compared to the control group. Data are representative of three independent experiments as mean ± SD.

**Figure 5 antioxidants-11-00221-f005:**
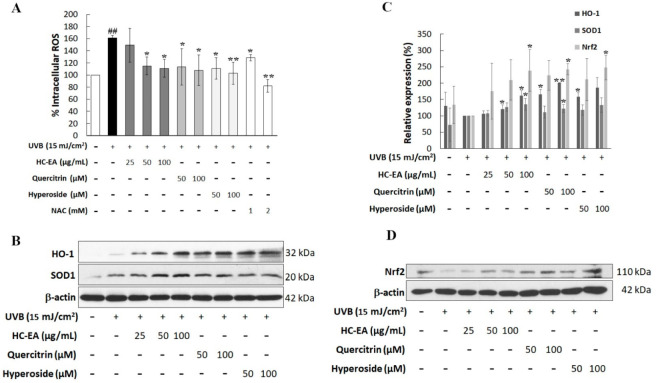
Antioxidant potential of HC-EA and its active compounds on UVB-induced HaCaT cells. (**A**) The production of intracellular ROS in HaCaT cells was measured at 2 h after UVB irradiation (15 mJ/cm^2^) by using DCF-DA dye, then the fluorescent intensity was determined by spectrofluorometer. The effect of HC-EA and its bioactive compounds on the expression of antioxidant proteins SOD and HO-1 (**B**) and transcription factor Nrf2 (**D**) were evaluated by Western blotting. Densitometric and statistical analysis of protein quantification data normalized to β-actin are presented as histogram (**C**). All experiments were performed at least three times. * *p* < 0.05, ** *p* < 0.01, as compared to the UVB-irradiated alone group. ## *p* < 0.01 as compared to the control group. Data are representative of three independent experiments as mean ± SD.

**Figure 6 antioxidants-11-00221-f006:**
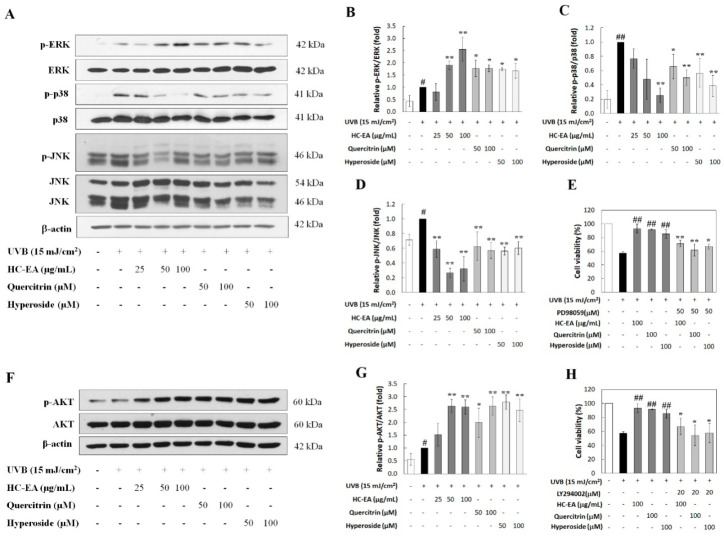
Effect of HC-EA and its active compounds on the modulation of MAPKs and Akt signaling pathway. HaCaT cells were pretreated with the indicated concentrations of HC-EA and its active compounds for 6 h. After UVB irradiation for 1 h, the cells were collected and subjected to investigation of the activation of MAPKs (**A**) and Akt (**F**) by Western blot analysis. Densitometric and statistical analysis were performed to determine the relative phosphorylation ratio of ERK (**B**), p-38 (**C**), JNK (**D**) and Akt (**G**). All experiments were performed at least three times. * *p* < 0.05, ** *p* < 0.01, as compared to the UVB-irradiated alone group. # *p* < 0.05, ## *p* < 0.01, as compared to the control group. Data are representative of three independent experiments as mean ± SD. To investigate the activation effect of ERK and Akt by HC-EA and its active compounds, which is responsible for photoprotection in UVB-induced cell damage, the cells were pretreated with HC-EA or its active compounds along with ERK inhibitor (PD98056) (**E**) or Akt inhibitor (LY294002) (**H**). After UVB irradiation for 24 h, cell viability was determined by SRB assay. All experiments were performed at least three times. * *p* < 0.05, ** *p* < 0.01, as compared to UVB-irradiated combination with the tested compounds group. ## *p* < 0.01 as compared to the UVB-irradiated alone group. Data are representative of three independent experiments as mean ± SD.

**Table 1 antioxidants-11-00221-t001:** Determination of total phenolic contents, total flavonoid contents, and phytochemical compounds in HC-EA. Data are representative of three independent experiments as mean ± SD.

Compounds	HC-EA
**Total Phenolic (mg Gallic Acid/g Extract)**	**718.71 ± 58.29**
Chlorogenic acid (mg/g Extract)	63.40 ± 1.88
**Total Flavonoid (mg Catechin/g Extract)**	**437.05 ± 21.01**
Hyperoside (mg/ g Extract)	230.80 ± 1.37
Quercitrin (mg/ g Extract)	286.91 ± 0.55

**Table 2 antioxidants-11-00221-t002:** ABTS and DPPH free radical activity of the HC-EA and its actives compounds were examined in the present study.

Compounds	DPPH Radical Scavenging Activity (IC_50_)	ABTS Radical Scavenging Activity (IC_50_)
Vitamin E (µg/mL)	36.1 ± 6.8	-
Trolox (µg/mL)	-	2.4 ± 0.3
HC-EA (µg/mL)	21.7 ± 1.4	4.5 ± 0.3
Quercitrin (μM)	70.7 ± 9.2	13.3 ± 4.0
Hyperoside (μM)	55.3 ± 1.2	12.2 ± 5.7

## Data Availability

Data is contained within the article and Appendix A.

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
