# Peer review of "Hyperoside and Quercitrin in Houttuynia cordata Extract Attenuate UVB-Induced Human Keratinocyte Cell Damage and Oxidative Stress via Modulation of MAPKs and Akt Signaling Pathway"

_antioxidants, 2022, doi:10.3390/antiox11020221_

Round 1
Reviewer 1 Report
The authors of "Hyperoside and Quercitrin in Houttuynia cordata Extract Attenuate UVB-Induced Human Keratinocyte Cell Damage and Oxidative Stress via Modulation of MAPKs and Akt Signaling Pathway" explored the impact of HC-EA and its natural components on their ability to protect HaCaT cells from harmful UVB light. The authors examined cell viability post exposure as well as a number of oxidative stress and signal transduction targets. This article was well written and was of great interest as UVB exposure is a rising concern. Below are a few minor points that arose during revisions.
1) In several figures the authors show that the components as well as HC-EA both reduce oxidative stress damage. The authors should comment on and discuss the mechanism for this. If you add the components together would you see a combinatorial or a synergistic response?
2) While gel images are very helpful to see trends, it is impossible to perform statistics with one set visible. In addition to showing a representative image, it is suggested that the images be analyzed and converted into a graph with statistical analysis carried out.
Author Response
Comment 1: “In several figures the authors show that the components as well as HC-EA both reduce oxidative stress damage. The authors should comment on and discuss the mechanism for this. If you add the components together would you see a combinatorial or a synergistic response”
Response: Per advice, we discussed the mechanism of quercitrin and hyperoside in HC-EA in discussion section, page 15, line 576-583.
Comment 2: “While gel images are very helpful to see trends, it is impossible to perform statistics with one set visible. In addition to showing a representative image, it is suggested that the images be analyzed”
Response: We quantified the band intensity of all western blot results and presented by histogram graph with statistical analysis as shown in Figure 3F, 4H, 5D, 6B-D, and 6G.
Reviewer 2 Report
The paper presented for review is made very carefully and reliably. The project idea and analysis were presented logically and based on the literature knowledge. I appreciate the authors' effort to prepare a good project and an excellent manuscript. I have only one remark that will not change the value of the results, but the analysis's correctness has to be changed. Statistical analysis must be performed based on the ANOVA test and selected post-hoc rather than the Student t-test.
Author Response
Comment: The paper presented for review is made very carefully and reliably. The project idea and analysis were presented logically and based on the literature knowledge. I appreciate the authors' effort to prepare a good project and an excellent manuscript. I have only one remark that will not change the value of the results, but the analysis's correctness has to be changed. Statistical analysis must be performed based on the ANOVA test and selected post-hoc rather than the Student t-test.
Response: We changed statistical analysis to be performed by one-way ANOVA with Dunnett’s test as described in Material and Methods section, page 5, line 223-225. We already checked and changed the statistical significance in all figures.
Reviewer 3 Report
The authors investigated the effect of Hyperoside and Quercitrin from Houttuynia cordata Extract At- to attenuate oxidative stress induced by UVB treatment on Human Keratinocyte Cell Some points: -Please add a setting with an antioxidant e.g., NAC to validate ROS induction. -Dichlorofluorescein diacetate was used to evaluate iintracellular ROS. The use of insensitive [C-369; 5-(and-6)-carboxy-2¢,7¢-dichlorofluorescein diacetate] fluorescent dyes was not used. The oxidation insensitive probe could be utilized to control for changes in uptake, ester cleavage, and efflux so that differences in fluorescence can definitively be attributed to changes in oxidation of the probe. -What about polyphenol bioavailability in keratinocytes? -All polyphenols possess notable reducibility properties. How can the authors rule out this interference into the outcome? That is, the eased oxidative stress could largely attribute to the reducibility properties.Author Response
Comment 1: “Please add a setting with an antioxidant e.g., NAC to validate ROS induction”
Response: Per advice, the effect of NAC on UVB-induced intracellular ROS was added into Figure 5A and in the result section page 10, line 364-365.
Comment 2: “Dichlorofluorescein diacetate was used to evaluate iintracellular ROS. The use of insensitive [C-369; 5-(and-6)-carboxy-2¢,7¢-dichlorofluorescein diacetate] fluorescent dyes was not used. The oxidation insensitive probe could be utilized to control for changes in uptake, ester cleavage, and efflux so that differences in fluorescence can definitively be attributed to changes in oxidation of the probe.”
Response: Per advice, we inserted the principle of DCF-DA Assay in Material and Methods section, page 4, line 180-186.
In this study, we used DCF-DA to determine intracellular ROS in keratinocytes instead of 5(6)-Carboxy-2′,7′-dichlorofluorescein diacetate (CDCF-DA) as described below.
“CDCF-DA is fluorescein-based indicator has significant advantages over previous fluorescein diacetate-based tracer systems in that it rapidly and efficiently diffuses into cells as a colorless, non-fluorescent probe until the two acetate groups are cleaved by intracellular esterases to yield the fluorescent fluorophore, 5-(and-6)-carboxy-2′,7′-dichlorofluorescein (CDCF) and becomes fluorescent when the dihydrofluorescein is oxidized to fluorescein by intracellular ROS. Moreover, CDCF-DA has exceedingly low sensitivity of intrinsic fluorescence yield at intracellular pH, and the 5′-and 6′-carboxy-ligands significantly further stabilize the internalized fluorescent signal enabling longer data-collection times. Moreover, CDCF is used as a fluorescent substrate/probe to study drug uptake and efflux mechanism in various of cells. CDCF is a substrate for MRP1 and MRP2 but not for P-gp, has been demonstrated to allow sensitive and specific detection of cellular MRP-related transport activity. The highly expression of MRP-1 and -2 on the cell membrane of HaCaT cells have been reported. Therefore, a few reports have been used CDCF-DA to study intracellular ROS in HaCaT cell according to expression of MRP proteins.”
Comment 3: What about polyphenol bioavailability in keratinocytes?
Response: Per advice, we added the polyphenol bioavailability in keratinocytes in Discussion section, page 15, line 584-604.
Comment 4: All polyphenols possess notable reducibility properties. How can the authors rule out this interference into the outcome? That is, the eased oxidative stress could largely attribute to the reducibility properties.
Response: We added the correlation between reducing power and antioxidant property of polyphenols in Discussion section, page 14, line 534-539.